# Eating Competence among Brazilian College Students

**DOI:** 10.3390/ijerph20043488

**Published:** 2023-02-16

**Authors:** Caroline Frois Boeira, Fabiana Lopes Nalon de Queiroz, Renata Puppin Zandonadi, Helena Beatriz Rower, Eduardo Yoshio Nakano, Ana Maria Pandolfo Feoli

**Affiliations:** 1Eating Behavior Group of the Psychology Postgraduate Program, Pontifical Catholic University of Rio Grande do Sul, Porto Alegre 90619-900, Brazil; 2Department of Nutrition, College of Health Sciences, Darcy Ribeiro University Campus, University of Brasília, Brasília 70910-900, Brazil; 3Department of Statistics, University of Brasília, Brasília 70910-900, Brazil

**Keywords:** eating behavior, eating competence, lifestyle, students’ health

## Abstract

Eating competence (EC) is a biopsychosocial concept related to a healthier eating pattern. Studies have shown that weight gain and body shape and weight dissatisfaction are typical among college students, contributing to low self-esteem, risky eating behaviors, and a propensity to develop eating disorders. As eating habits are determinant factors for food choices that can be modified by eating behavior, this study aimed to evaluate EC in college students from Brazil, using the Brazilian version of the EC Satter Inventory (ecSI2.0™BR), and EC’s association with health data. This cross-sectional study was conducted using an online survey spread using the snowball method. The self-report instrument was divided into three parts (socioeconomic and demographic data; health data; and ecSI2.0™BR). Recruitment took place through social networks, and 593 students from public and private universities from all five regions of Brazil participated in the survey. The EC average was 29.46 ± 8.67, and 46.2% of the sample were considered competent eaters. Total EC did not differ among gender and Brazilian region. Younger participants (up to 20 y/o) presented higher scores for total EC, contextual skills and food acceptance. The total EC and contextual skills of health sciences students did not differ from those of students in other areas, except for agricultural sciences in which students had lower total EC. Obese individuals and the participants who perceived themselves to be overweight had low scores for EC. This study confirmed the hypothesis that college students have low EC, causing worse health outcomes related to BMI, perceived body weight, and occurrence of hypertension and dyslipidemia.

## 1. Introduction

Eating is necessary for survival, but goes far beyond that. It involves issues that are external to the individual, such as social, economic, political, and cultural factors, in addition to being associated with individual factors that influence food choices and nutrition. Thus, regarding individual choices, eating habits can be modified through eating behavior changes and improving food choices. Among behavioral concepts related to nutrition, eating competence (EC) is an approach that has been associated with a healthier eating pattern [1].

The EC model, proposed by Satter, is an approach based on the effectiveness of the following biopsychosocial processes of eating: hunger as a need for survival, appetite and reward-seeking, and the natural tendency to maintain body weight [2]. Eating competence is not based on specific nutrients, but it is characterized by flexibility, optimism, confidence, and comfort regarding food choices [3,4]. EC can be measured using the Satter Eating Competence Inventory (ecSI2.0™) [4], a questionnaire developed for non-pregnant and non-lactating adults in the US population, structured with the same EC self-regulated components. This instrument was translated and validated to the Brazilian version of the EC Satter Inventory (ecSI2.0™BR) [5], to be used for the Brazilian adult population under the concession of the NEEDs Center [6].

EC comprises four elements: eating attitude, food acceptance, internal regulation, and contextual skills. Eating attitude refers to a flexible, positive, and calm attitude towards eating along with the individual’s internal and external experiences, connected to the expectation and anticipation of eating something palatable. Food acceptance describes the individual’s nutrition based on personal experiences, and is determined by the internal motivation to eat a variety of foods with pleasure, respecting food preferences while still maintaining an aptitude for trying new foods. Internal regulation refers to attention to internal hunger cues and satiety signals as regulators of the amount to be ingested [7]. Contextual skills deal with one’s own food management, including food acquisition and meal planning, as well as being able to eat preferred foods in adequate amounts and at structured times [2]. Thus, “competent eaters” are those who allow themselves to eat foods that bring satisfaction (respecting their internal regulation of hunger and satiety) and are open to trying new foods, planning and organizing their meals, and eating a wide variety of foods [3].

EC positively affects adults’ health [8], especially in preventing cardiovascular diseases [2,3,4,9], through a higher-quality diet (rich in nutrients, minerals, and fibers) [10]. It also positively impacts general health indicators, such as higher high-density lipoprotein (HDL) cholesterol levels, a lower body mass index (BMI), and lower blood glucose concentration [11]. In addition, EC is associated with increased consumption of fruits and vegetables [10], greater practice of physical activity [12], better organization of the food context [13], better sleep quality [14,15], and a positive impact on children’s eating and nutrition [9,14,16,17].

Weight gain, body dissatisfaction, and weight dissatisfaction are common among young adult college students [3]. In addition to many of them being obese or overweight, body dissatisfaction causes low self-esteem, risky eating behaviors, and a predisposition to eating disorders. Furthermore, the pursuit of weight loss caused by the concern about weight, and not exactly the weight itself, can create a diet mentality, which is a set of beliefs regarding food and the body, as well as beliefs regarding the dichotomization of permitted and prohibited foods [18], ultimately causing changes in lifestyle and health outcomes [3].

There are still no published studies in Brazil describing EC measurements in college students. Studies conducted with college students in the United States have linked a lack of EC with greater body weight dissatisfaction and higher BMI compared to those who have achieved EC [19]. Additionally, among college students, EC is gender-related, with women showing lower EC than men [3]. Another important association concerns eating disorders in college students; those who report current or past eating disorders obtain lower EC scores than those who have never had eating disorders [19]. Thus, this study aimed to evaluate the EC of college students in Brazil through the ecSI2.0™BR and its association with health data. Since EC is a concept related to a healthier eating pattern, we hypothesized that college students with low EC scores would have worse health outcomes.

## 2. Materials and Methods

### 2.1. Ethics and Study Design

This quantitative cross-sectional study was performed from 31 August 2022 to 4 November 2022, using an online survey spread by the snowball method. The instrument was applied nationwide through the Qualtrics survey software. A self-reported instrument consisted of three parts: (i) Socioeconomic and demographic data; (ii) health data; and (iii) the “ecSI2.0™BR”. This study was approved by the Research Ethics Committee of the Pontifical Catholic University of Rio Grande do Sul under CAAE number 62597922.2.0000.5336. All participants signed an informed consent form containing more information about the project and contact information for questions.

### 2.2. Participants

The inclusion criteria were as follows: (1) undergraduate program enrollment at universities, university centers, or public or private colleges throughout Brazil; (2) ≥18 y/o; (3) participation agreement confirming the reading of the informed consent form; and (4) no pregnancy. The convenience sample was collected online, through disclosure via Instagram, WhatsApp, and email from university coordinators, university centers, and public and private colleges, and full completion of the questionnaire. The digital platform used to apply the questionnaire was the Qualtrics survey software.

We calculated the minimum sample size that predicted a minimum of 425 students throughout Brazil, respecting a minimum sample for each region of the country based on the proportion of students enrolled in these regions, according to data from National Institute for Educational Studies and Research Anísio Teixeira (Inep) [20], as follows: North = 28, Northeast = 96, Midwest = 38, Southeast = 199 and South = 64.

Participation was anonymous, but subjects could share their email address at the end to receive feedback on the questionnaire score. Participants could leave the survey anytime, so their answers were not considered.

### 2.3. Instrument Application

The questionnaire comprised 37 items divided into three sections. The first part of the survey included 12 items to obtain demographic data: age, sex, place of birth, city and state of residence, course, semester, institution, course modality and class shift, and current occupation. In the second part, participants answered questions on general health, including self-reported weight and height and perceived body weight. Participants’ BMI was calculated from the self-reported weight and height, establishing reference values as underweight for a BMI less than 18.5 kg/m^2^, healthy weight range between 18.5 kg/m^2^ and 24.9 kg/m^2^, overweight between 25 kg/m^2^ and 29.9 kg/m^2^ and obesity above 30 kg/m^2^, according to the classification adopted by the World Health Organization (WHO) [21]. Finally, in the third part, participants accessed the ecSI2.0™BR [5]. The instrument is available online at https://www.needscenter.org/ and was accessed after NEEDs Center approval for this study (on 15 June 2022) [6]. The ecSI2.0™BR is a self-reporting tool comprised 16 items: six related to eating attitude, three on food acceptance, two on internal regulation of food intake, and five on contextual skills [22]. Each question has five possible answers: never = 0; rarely = 0; sometimes = 1; frequently = 2 and always = 3. The final score ranges from 0 to 48, with scores equal to or above 32 being considered a competent eater.

### 2.4. Statistical Analysis

The ecSI2.0™BR scores and their four components were described using their means and standard deviation (SD). A Student’s *t*-test and analysis of variance (ANOVA) followed by Tukey’s test were used to compare the instrument scores with the variables of interest. The results of categorized EC (ecSI2.0™BR ≥ 32) were described using frequencies and percentages, and their association with the variables of interest was verified using Pearson chi-squared test and Fisher’s exact test. All tests considered two-tailed hypotheses and a significance level of 5%. Analyses were performed using the IBM SPSS program (IBM SPSS Statistics for Windows, IBM Corp., Armonk, NY, USA) version 22.

## 3. Results

A total of 827 answers were recorded. From them, 210 that were incomplete and 24 that did not meet the inclusion criteria (non-acceptance of the informed consent form, no undergraduate program enrollment, pregnancy) were excluded, computing 593 valid answers. The sample was primarily women (*n* = 448; 75%), ≥21 years of age (*n* = 450), studying in the area of health science (*n* = 261), and from the southern region (*n* = 245) (Table 1).

The total EC mean score was 29.46 ± 8.67, where 46.2% of students were considered competent eaters (ecSI2.0™BR ≥ 32). There was no significant difference between females and males in the total score, only in the contextual skills (*p* = 0.007) and food acceptance (*p* = 0.042) components, where women showed higher scores. Participants aged up to 20 years had higher scores on the ecSI2.0™BR (*p* = 0.014) and the contextual skills (*p* = 0.026) and food acceptance (*p* = 0.045) components. Regarding the area of study, health science students are more competent eaters than exact and earth sciences students (*p* = 0.007).

College students in the hybrid modality were considered more competent eaters than students in the online learning modality (*p* = 0.033), and also scored higher in total EC (*p* = 0.034) and in the contextual skills component (*p* = 0.01). There was no difference regarding the students’ current occupations.

Regarding general health data, results show that obese students had a lower mean EC compared to participants considered underweight, overweight, and normal weight (*p* < 0.001). The same was observed in the contextual skills (*p* < 0.001) and eating attitude (*p* < 0.001) components. Obese participants were also considered less competent eaters than participants classified as underweight and within the healthy weight range (*p* < 0.001). Five participants did not report weight and height correctly, and it was not possible to calculate BMI.

Regarding perceived body weight, participants who perceived themselves as feeling good about their current weight or underweight were considered more competent eaters and scored higher in total EC than those who perceived themselves as overweight (*p* < 0.001).

There was no significant difference in scores of the participants who have diabetes. However, there was a difference in those who reported hypertension (*p* = 0.034) and dyslipidemia (*p* = 0.043), who showed lower mean scores. Participants with hypertension also had lower scores in the contextual skills component (*p* = 0.004), and those with dyslipidemia had lower scores in the eating attitude (*p* = 0.005) and internal regulation (*p* = 0.007) components.

## 4. Discussion

The results showed that less than half of the students (46.2%) were considered competent eaters, evidence that is different from that found in a study with sexual and gender minority US college students in which 52% were considered competent eaters [23], but that is similar to previous studies conducted with US college students in general [3,15,19,24]. Three of these studies were conducted with college students in general [3,15,24] and one with college students of the health science area who were enrolled in an introductory nutrition course [19], with a biased sample, according to the authors, of students of the health area. In contrast, despite our study mainly comprising students in the area of health science, there was no significant difference in the percentage of college students considered competent eaters between the other six areas, except exact and earth sciences.

College students aged up to 20 y/o had higher scores on the general scale, contextual skills, and food acceptance components. This indicates possible interference from the environment, since there is a high probability of this age group still living with parents who would be responsible for food organization in a general context, and having a greater variety of foods. Studies conducted with young adults up to 23 years of age have demonstrated parental influence on their eating habits, wherein those who live with their parents have a higher consumption of fruits and vegetables [25]; they also receive greater family support in meal preparation during test periods, and better habits overall when parents are in control of food organization [26].

In our study, there were no significant differences between sexes in the total score or frequency regarding EC, as was found in a study with the Brazilian adult population [27]. However, this differs from the findings of other studies [3,8,19,24], which describe men as more competent eaters than women. Despite this, according to Lohse et al. [4], compared with other characteristics related to EC such as food preference and physical activity), sex and BMI may not be predictors of EC. Seeking to explain this relationship, two studies demonstrate that EC seems to vary according to age [19,28]. In one of the studies, the sample comprised college students aged 18 to 25 years, and the EC was higher in men than in women [19]. In the other study composed of adolescents aged 10 to 17 years EC was higher in girls than in boys [28]. Another important association involves income, where low ecSI2.0™ scores of women were related to dissatisfaction with body weight, eating disorders, and a greater propensity to overeat due to external emotional factors [29]. Another study evaluated the relationship between orthorexia nervosa (ON) and EC among college students. The authors showed an association between ON and EC in both sexes, with lower positive EC scores in eating attitude and internal regulation. Female participants’ levels of orthorexic attitudes were higher than in male participants [30].

Obese students, according to their BMI, were considered less competent eaters and had lower scores in the EC mean in our study. This finding is consistent with other studies on college students [15,24] and the adult population in general [3,4,14,22,29,31]. In addition to a lower overall score on the scale, obese individuals also scored lower on their eating attitude, contextual skills, and internal regulation components. This indicates that these students may not have a positive attitude towards food as well as not having good organization of meals; in addition, they may not pay attention to hunger cues and satiety signs which in turn would be a natural regulator of the amount to be ingested [2].

Another important piece of data found in this study involves the participants’ perceived body weight, as it was observed that some students who perceived themselves to be overweight were neither overweight nor obese. This finding relates to what was exposed by Strauss [32], and Avalos and Tylka [33], who have demonstrated that perceived body weight is a better predictor of dieting behaviors than weight itself. This dieting behavior is mainly characterized by weight loss attempts, which modify eating behavior, reflecting in EC and showing that college students with the desire to lose weight tend to be non-competent eaters [3]. This study corroborates these findings, where participants who perceived themselves to feel good about their current weight or as underweight were considered more competent eaters than those who perceived themselves as overweight.

Additionally, regarding weight perception, those students who perceived themselves as underweight and felt good about their current weight scored higher in the eating attitude and internal regulation components than participants who believed they were overweight. These also scored lower in the contextual skills component than participants who feel good about their weight.

Of the college students who declared having diabetes and hypertension, only 12.5% were considered competent eaters, and of the participants with dyslipidemia, 32.6%. With these questions, it was expected that patients with chronic non-communicable diseases, which directly affect cardiovascular health, would be more competent in food choices as a form of treatment for the disease and change of habits after the diagnosis. As seen before, EC is related to preventing cardiovascular problems [9,10,11]; therefore, approaching this population with a focus on EC-based nutritional education can benefit these individuals’ quality of life [9]. Furthermore, non-dietary interventions that target behavioral change may also be beneficial for improving EC in other populations [34].

One strength of this study is that answers were obtained from all regions of the country and from all areas of study. In addition, this work highlights aspects of the diet of university students in the country, providing information to increase awareness and plan interventions in this population to improve their quality of life. However, this study did not collect other variables of possible interest that could interfere with EC and allow for more in-depth analysis, for example, through questions that qualitatively addressed food, desire for weight loss and body satisfaction.

## 5. Conclusions

The study aimed to evaluate EC in college students in Brazil and its association with health data, this being the first study on EC conducted with this population in the country. EC is related to a positive attitude towards food and healthier eating habits. This study confirmed the hypothesis that college students have low EC and worse health outcomes related to BMI, perceived body weight, and occurrence of hypertension and dyslipidemia. Obese individuals and the participants who perceived themselves to be overweight had low scores for EC. Participants with hypertension or dyslipidemia presented lower EC total scores than those that do not have these chronic diseases, but there was no significant difference in the scores of participants who have diabetes. Therefore, EC is directly related to BMI, perceived body weight, and occurrence of hypertension and dyslipidemia.

Knowing the health reality of college students and developing educational strategies for health promotion to make them competent eaters can promote better health outcomes, mainly considering non-communicable chronic diseases. Further longitudinal studies with this population are necessary to understand if university life can influence EC.

## Figures and Tables

**Table 1 ijerph-20-03488-t001:** ecSI2.0™BR sub-scores categorized by variables (*n* = 593, Brazil).

	Eating Attitude	Food Acceptance	Internal Regulation	Contextual skills	Total	ecSI2.0™BR ≥ 32
	Mean (SD)	Mean (SD)	Mean (SD)	Mean (SD)	Mean (SD)	Freq. (%)
General (*n* = 593)	11.78 (3.87)	4.82 (2.49)	3.99 (1.46)	8.87 (3.63)	29.46 (8.67)	274 (46.2%)
Sex						
Female (*n* = 448)	11.63 (3.94) ^a^	4.94 (2.43) ^b^	3.99 (1.45) ^a^	9.12 (3.50) ^b^	29.67 (8.54) ^a^	216 (48.2%) ^a^
Male (*n* = 145)	12.24 (3.61) ^a^	4.46 (2.62) ^a^	3.99 (1.51) ^a^	8.12 (3.94) ^a^	28.81 (9.07) ^a^	58 (40.0%) ^a^
*p*	0.086 ^1^	0.042 ^1^	0.963 ^1^	0.007 ^1^	0.296 ^1^	0.085 ^3^
Age						
Up to 20 years old (*n* = 143)	12.31 (3.94) ^a^	5.18 (2.54) ^b^	4.06 (1.49) ^a^	9.46 (3.59) ^b^	31.01 (8.74) ^b^	81 (56.6%) ^b^
21 years or older (*n* = 450)	11.62 (3.83) ^a^	4.70 (2.46) ^a^	3.96 (1.45) ^a^	8.68 (3.63) ^a^	28.97 (8.60) ^a^	193 (42.9%) ^a^
*p*	0.062 ^1^	0.045 ^1^	0.483 ^1^	0.026 ^1^	0.014 ^1^	0.004 ^3^
Region						
North (*n* = 49)	11.29 (4.12) ^a^	4.12 (2.53) ^a^	4.29 (1.43) ^a^	8.71 (3.62) ^a^	28.41 (9.04) ^a^	24 (49.0%) ^a^
Northeast (*n* = 104)	11.58 (3.91) ^a^	4.18 (2.32) ^ab^	4.14 (1.47) ^a^	8.54 (3.84) ^a^	28.44 (8.86) ^a^	46 (44.2%) ^a^
Southeast (*n* = 152)	11.91 (3.91) ^a^	5.26 (2.37) ^b^	0.91 (1.50) ^a^	9.23 (3.58) ^a^	30.32 (8.49) ^a^	73 (48.0%) ^a^
South (*n* = 245)	11.98 (3.81) ^a^	5.03 (2.58) ^ab^	3.89 (1.42) ^a^	9.01 (3.64) ^a^	29.91 (8.73) ^a^	117 (47.8%) ^a^
Midwest (*n* = 43)	11.26 (3.73) ^a^	4.40 (2.22) ^ab^	4.09 (1.52) ^a^	7.79 (3.20) ^a^	27.53 (7.78) ^a^	14 (32.6%) ^a^
*p*	0.608 ^2^	0.001 ^2^	0.294 ^2^	0.158 ^2^	0.170 ^2^	0.409 ^3^
Area of study						
Exact and earth sciences (*n* = 44)	10.66 (3.39) ^ab^	3.80 (1.91) ^a^	4.00 (1.48) ^a^	7.34 (3.73) ^ab^	25.80 (7.28) ^ab^	10 (22.7%) ^a^
Biological sciences (*n* = 14)	10.29 (4.08) ^a^	5.43 (2.17) ^ab^	3.93 (1.82) ^a^	8.86 (3.96) ^ab^	28.50 (8.79) ^ab^	6 (42.9%) ^ab^
Engineering (*n* = 37)	12.35 (3.98) ^ab^	4.84 (2.46) ^ab^	3.81 (1.68) ^a^	8.73 (3.24) ^ab^	29.73 (8.48) ^ab^	18 (48.6%) ^ab^
Health science (*n* = 261)	12.25 (3.70) ^ab^	5.16 (2.54) ^b^	4.00 (1.37) ^a^	9.55 (3.41) ^b^	30.96 (8.40) ^b^	140 (53.6%) ^b^
Agricultural science (*n* = 22)	10.41 (4.54) ^ab^	4.27 (2.47) ^ab^	3.68 (1.49) ^a^	6.45 (4.39) ^a^	24.82 (10.27) ^a^	7 (31.8%) ^ab^
Applied social sciences (*n* = 135)	11.56 (3.97) ^ab^	4.84 (2.47) ^ab^	4.09 (1.50) ^a^	8.40 (3.56) ^ab^	28.89 (8.51) ^ab^	55 (40.7%) ^ab^
Human science (*n* = 63)	11.19 (4.08) ^ab^	4.44 (2.49) ^ab^	3.87 (1.55) ^a^	8.76 (3.68) ^ab^	28.27 (9.45) ^ab^	29 (46.0%) ^ab^
Linguistics, letters and arts (*n* = 17)	13.24 (3.33) ^b^	3.59 (2.40) ^ab^	4.29 (1.40) ^a^	10.00 (4.09) ^b^	31.12 (8.01) ^b^	9 (52.9%) ^ab^
*p*	0.014 ^2^	0.005 ^2^	0.855 ^2^	<0.001 ^2^	0.001 ^2^	0.007 ^3^
Course modality						
On-site (*n* = 554)	11.78 (3.89) ^a^	4.81 (2.47) ^a^	3.99 (1.45) ^a^	8.83 (3.61) ^ab^	29.41 (8.67) ^ab^	257 (46.4%) ^ab^
Hybrid (*n* = 23)	12.43 (2.92) ^a^	5.78 (2.63) ^a^	4.09 (1.59) ^a^	10.83 (2.81) ^b^	33.13 (7.65) ^b^	14 (60.9%) ^b^
Online learning (*n* = 16)	10.81 (4.21) ^a^	3.94 (2.54) ^a^	3.75 (1.57) ^a^	7.44 (4.59) ^a^	25.94 (8.94) ^a^	3 (18.8%) ^a^
*p*	0.437 ^2^	0.064 ^2^	0.076 ^2^	0.010 ^2^	0.034 ^2^	0.033 ^3^
Class shift						
Morning (*n* = 110)	12.07 (3.99) ^a^	5.00 (2.60) ^a^	4.07 (1.44) ^a^	9.28 (3.60) ^a^	30.43 (8.41) ^a^	51 (46.4%) ^a^
Night (*n* = 185)	11.62 (3.88) ^a^	4.73 (2.46) ^a^	3.84 (1.52) ^a^	8.63 (3.75) ^a^	28.82 (9.09) ^a^	83 (44.9%) ^a^
Online learning (*n* = 17)	10.88 (4.09) ^a^	4.35 (2.78) ^a^	3.65 (1.46) ^a^	8.18 (4.30) ^a^	27.06 (9.06) ^a^	4 (23.5%) ^a^
Other (*n* = 281)	11.83 (3.81) ^a^	4.84 (2.44) ^a^	4.07 (1.42) ^a^	8.91 (3.53) ^a^	29.65 (8.45) ^a^	136 (48.4%) ^a^
*p*	0.588 ^2^	0.699 ^2^	0.237 ^2^	0.415 ^2^	0.281 ^2^	0.241 ^3^
Current occupation						
Full-time student (*n* = 255)	11.63 (3.86) ^a^	4.81 (2.39) ^a^	4.03 (1.42) ^a^	8.93 (3.64) ^a^	29.40 (8.38) ^a^	114 (44.7%) ^a^
Carrying out a paid activity (*n* = 233)	11.91 (3.98) ^a^	4.76 (2.53) ^a^	3.95 (1.53) ^a^	8.78 (3.78) ^a^	29.40 (9.19) ^a^	104 (44.6%) ^a^
Carrying out compulsory internship (*n* = 105)	11.86 (3.67) ^a^	4.97 (2.61) ^a^	3.97 (1.40) ^a^	8.95 (3.31) ^a^	29.75 (8.26) ^a^	56 (53.3%) ^a^
*p*	0.706 ^2^	0.768 ^2^	0.816 ^2^	0.876 ^2^	0.931 ^2^	0.271 ^3^
BMI (kg/m^2^) *						
Underweight < 18.5 (*n* = 33)	12.94 (3.71) ^b^	3.85 (2.53) ^a^	4.39 (1.48) ^b^	9.03 (4.00) ^b^	30.21 (9.02) ^b^	17 (51.5%) ^b^
Healthy weight range 18.5–24.9 (*n* = 367)	12.24 (3.70) ^b^	5.05 (2.45) ^b^	4.13 (1.42) ^b^	9.28 (3.40) ^b^	30.70 (8.00) ^b^	188 (51.2%) ^b^
Overweight 25–29.9 (*n* = 128)	11.38 (3.68) ^b^	4.67 (2.45) ^ab^	3.79 (1.43) ^ab^	8.74 (3.71) ^b^	28.59 (8.58) ^b^	54 (42.2%) ^ab^
Obesity ≥ 30 (*n* = 60)	9.32 (4.26) ^a^	4.28 (2.62) ^ab^	3.30 (1.53) ^a^	6.48 (3.84) ^a^	23.38 (10.12) ^a^	13 (21.7%) ^a^
*p*	<0.001 ^2^	0.009 ^2^	<0.001 ^2^	<0.001 ^2^	<0.001 ^2^	<0.001 ^3^
Perceived body weight						
See themselves as overweight (*n* = 219)	10.09 (4.00) ^b^	4.64 (2.52) ^a^	3.40 (1.50) ^a^	7.84 (3.78) ^a^	25.98 (9.17) ^a^	62 (28.3%) ^a^
See themselves as underweight (*n* = 49)	11.96 (3.97) ^a^	4.37 (2.57) ^a^	4.47 (1.42) ^b^	8.73 (3.68) ^ab^	29.53 (9.36) ^b^	24 (49.0%) ^b^
Feels good about current weight (*n* = 325)	12.90 (3.33) ^a^	5.01 (2.44) ^a^	4.31 (1.31) ^b^	9.58 (3.36) ^b^	31.80 (7.36) ^b^	188 (57.8%) ^b^
*p*	<0.001 ^2^	0.103 ^2^	<0.001 ^2^	<0.001 ^2^	<0.001 ^2^	<0.001 ^3^
Do you have diabetes?						
Yes (*n* = 8)	10.63 (4.53) ^a^	4.38 (2.72) ^a^	3.13 (0.83) ^a^	8.25 (2.43) ^a^	26.38 (5.01) ^a^	1 (12.5%) ^a^
No (*n* = 585)	11.80 (3.86) ^a^	4.83 (2.48) ^a^	4.00 (1.46) ^a^	8.88 (3.65) ^a^	29.50 (8.71) ^a^	273 (46.7%) ^a^
*p*	0.395 ^1^	0.611 ^1^	0.092 ^1^	0.627 ^1^	0.311 ^1^	0.075 ^4^
Do you have high blood pressure (hypertension)?						
Yes (*n* = 8)	10.88 (5.03) ^a^	3.38 (2.67) ^a^	3.50 (1.20) ^a^	5.25 (3.28) ^a^	23.00 (8.68) ^a^	1 (12.5%) ^a^
No (*n* = 585)	11.79 (3.85) ^a^	4.84 (2.48) ^a^	3.99 (1.46) ^a^	8.92 (3.62) ^b^	29.55 (8.65) ^b^	273 (46.7%) ^a^
*p*	0.504 ^1^	0.098 ^1^	0.342 ^1^	0.004 ^1^	0.034 ^1^	0.075 ^4^
Do you have unhealthy blood fat levels (dyslipidemia)?						
Yes (*n* = 49)	10.31 (4.27) ^a^	5.31 (2.52) ^a^	3.45 (1.57) ^a^	8.00 (3.59) ^a^	27.06 (9.06) ^a^	16 (32.7%) ^a^
No (*n* = 544)	11.92 (3.81) ^b^	4.78 (2.48) ^a^	4.04 (1.44) ^b^	8.95 (3.63) ^a^	29.68 (8.61) ^b^	258 (47.4%) ^b^
*p*	0.005 ^1^	0.153 ^1^	0.007 ^1^	0.080 ^1^	0.043 ^1^	0.047 ^3^

* Five participants do not have BMI information; ^1^ Student’s *t*-test; ^2^ ANOVA with Tukey’s test; ^3^ Pearson chi-squared test; ^4^ Fisher’s exact test; Groups with equal letters do not differ statistically.

## Data Availability

Not applicable.

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
