# Peer review of "Eating Competence among Brazilian College Students"

_ijerph, 2023, doi:10.3390/ijerph20043488_

Round 1

Reviewer 1 Report

This manuscript deals with the important topic of eating competence in a vital section of the community of college students. The manuscript is well prepared. However, some concerns need to be fixed as follows:

1.      Line 66: the paragraph begins with studies. Thus, more than one reference should be present at the end of the paragraph. Also, “show” should be “showed”.

2.      Line 73-74: give references confirming these side effects in young college students.

3.      Add a subheading entitled “statistical analysis” for the paragraph “lines 129-136”.

4.      It is not preferable to begin sentences with abbreviations like EC in line 44.

5.      There is a problem with using abbreviations throughout the manuscript. The full term should be mentioned first with the abbreviation between paresis then the abbreviations should be exclusively used throughout the manuscript. E.g., eating competence has been abbreviated as EC then the full term has repeated again in lines 88 and 176, Also, in line 68 HDL should be given as high-density lipoprotein. Such errors have been repeated for many abbreviations throughout the manuscript.

Author Response

Reviewer #1:

This manuscript deals with the important topic of eating competence in a vital section of the community of college students. The manuscript is well prepared. However, some concerns need to be fixed as follows:

  1. Line 66: the paragraph begins with studies. Thus, more than one reference should be present at the end of the paragraph. Also, “show” should be “showed”.

R: Thank you for your comment. We added the references and changed the sentence.

  1. Line 73-74: give references confirming these side effects in young college students.

R: Thank you for your comment. The reference has been added.

  1. Add a subheading entitled “statistical analysis” for the paragraph “lines 129-136”.

R: Thank you for your comment. The subheading was added.

  1. It is not preferable to begin sentences with abbreviations like EC in line 44.

R: Thank you for your comment. It was changed accordingly in lines 44 and 52.

  1. There is a problem with using abbreviations throughout the manuscript. The full term should be mentioned first with the abbreviation between paresis then the abbreviations should be exclusively used throughout the manuscript. E.g., eating competence has been abbreviated as EC then the full term has repeated again in lines 88 and 176, Also, in line 68 HDL should be given as high-density lipoprotein. Such errors have been repeated for many abbreviations throughout the manuscript.

R: We carefully read the manuscript and adjusted all these issues. Thank you for your valuable comments on improving the quality of our manuscript!

Reviewer 2 Report

Eating Competence Among Brazilian College Students

Comments:

Kindly define the age limit of young adult students in Abstract.

Please mention the aim of your study in abstract.

Add conclusive line in the end of abstract.

Mention proper unit of Eating competence. Need space in unit.

You said EC  is a healthier eating pattern but in the last of introduction you said low EC have the worst health outcomes. Kindly explain it carefully.

Please add rationale of your study to highlight the reasoning of this article in the end of introduction section.

Please add sub headings in the materials and methods and please elaborate your methods briefly.

Article length is too short it will be better to add some more parameters.

Please you should use easy and simple language, don’t use harsh or difficult.

Some lines repeatedly kindly read whole article and erase that lines.

Please elaborate conclusion. Also mention relation between diabetes and eating competence in conclusion.

Why you discuss about covid 19 in your manuscript?

Recheck the all references. Add recent last five years references.

Author Response

Reviewer #2:

  1. Kindly define the age limit of young adult students in Abstract.

R: Thank you for your comment. The term was exchanged for “college students”.

  1. Please mention the aim of your study in abstract.

R: Thank you for your comment. The objective is placed in lines 16 to 18 of the abstract.

  1. Add conclusive line in the end of abstract.

R: Thank you for your comment. It was added in the abstract accordingly.

  1. Mention proper unit of Eating competence. Need space in unit.

R: Thank you for your comment, but Eating Competence do not have a unit of measurement; below some references that also used the instrument.

Queiroz FLN, Nakano EY, Botelho RBA, Ginani VC, Cançado ALF, Zandonadi RP. Eating Competence Associated with Food Consumption and Health Outcomes among Brazilian Adult Population. Nutrients. 2020 Oct 21;12(10):3218. doi: 10.3390/nu12103218. PMID: 33096760; PMCID: PMC7589896.

Queiroz FLN, Nakano EY, Cortez Ginani V, Botelho RBA, Araújo WMC, Zandonadi RP. Eating Competence among a Select Sample of Brazilian Adults: Translation and Reproducibility Analyses of the Satter Eating Competence Inventory. Nutrients. 2020 Jul 19;12(7):2145. doi: 10.3390/nu12072145. PMID: 32707639; PMCID: PMC7400831.

Queiroz FLN, Nakano EY, Botelho RBA, Ginani VC, Raposo A, Zandonadi RP. Eating Competence among Brazilian Adults: A Comparison between before and during the COVID-19 Pandemic. Foods. 2021 Aug 26;10(9):2001. doi: 10.3390/foods10092001. PMID: 34574111; PMCID: PMC8468240.

  1. You said EC  is a healthier eating pattern but in the last of introduction you said low EC have the worst health outcomes. Kindly explain it carefully.

R: We apologize if we were not clear. EC is not an eating pattern but a biopsychosocial concept related to a healthier eating pattern. Individuals with low EC scores tend to have the worst health outcomes. We changed the sentence to turn it clear.

  1. Please add rationale of your study to highlight the reasoning of this article in the end of introduction section.

R: Thank you for your comment. Some adjustments were made to clarify the study rationale.

  1. Please add sub headings in the materials and methods and please elaborate your methods briefly.

R: Thank you for your comment. We added subheadings accordingly.

  1. Article length is too short it will be better to add some more parameters.

R: Thank you for your comment. Some data and information has been input.

  1. Please you should use easy and simple language, don’t use harsh or difficult.

R: Thank you for your comment. The manuscript was revised in language.

  1. Some lines repeatedly kindly read whole article and erase that lines.

R: Thank you for your comment. We carefully read the manuscript and corrected it.

  1. Please elaborate conclusion. Also mention relation between diabetes and eating competence in conclusion.

R: Thank you for your comment. We’ve elaborated conclusion with some more information.

  1. Why you discuss about covid 19 in your manuscript?

R: Thank you for your comment. We’ve decided to withdraw that information.

  1. Recheck the all references. Add recent last five years references.

R: Thank you for your comment. The references were reviewed and updated.

Reviewer 3 Report

Dear Authors,

 Thank you for giving me the opportunity to review this manuscript entitled: "Eating Competence Among Brazilian College Students  ", to be considered for publication at IJERPH. Below you will find my comments and suggestions, which I believe can improve the quality of the final document. I really find this manuscript interesting?

Abstract: The information is accurate. Perhaps less methodological detail could have been provided in this section.

In the Reviewer's opinion, not all of the information placed on lines 18-22 is relevant at this stage.

I would suggest a rewording of the text:

Instead of: This cross-sectional study was performed using an online survey spread by snow-ball method and the instrument was applied nationwide through the Qualtrics survey software  from August 31, 2022 to November 4, 2022. A self-reported instrument (consisting of 37 questions) was divided into three parts: (i) Socioeconomic and demographic data; (ii) health data; and (iii) the  ecSI2.0™BR. The recruitment occurred through social networks and 593 college students from public and private Universities in all five Brazilian regions participated in the study.

Rather

This cross-sectional study was conducted using an online survey spread using the snow-ball method. The self-report instrument was divided into three parts.

Recruitment took place through social networks and 593 students from public and private universities in all five countries. lic and private universities in all five regions of Brazil participated in the survey.

Details removed should be included in the Materials and Methods section

Introduction: The introduction provides a solid study necessary to understand the subject. However, I think the authors should have cited more recent publications from the last five or even three years on eating competence. In fact, of the 28 citations, only 7 are from the last five years. This seems all the more surprising as the adaptation for the evaluation of the Satter method has only taken place for the Spanish-speaking countries, including Brazil in the year 2020.

Materials and method:

This section has been very well prepared by the authors. Nevertheless, I have a few comments.

Lines 103, please add information from abstract ”This cross-sectional study was performed using an online survey spread by snow-ball method and the instrument was applied nationwide through the Qualtrics survey software  from August 31, 2022 to November 4, 2022. A self-reported instrument (consisting of 37 questions)  was divided into three parts: (i) Socioeconomic and demographic data; (ii) health data; and (iii) the  ecSI2.0™BR”.

Line 104 -  In this section the number of participants (425) differs from the abstract and results (593), please correct.

Results: Nothing to say here. The information is accurate

Disscusion: The discussion was properly conducted, however, following on from comments made earlier, I miss the reference to recent research. In the last 5 years, in addition to the papers already cited by the authors of course, papers have been published with the following doi numbers; doi: 10.3390/nu12010104., doi: 10.3390/nu13114030. - diabetes,

Predictors of nutritional competence such as stress can also be referred to doi: The discussion has been conducted properly, but with reference to the comments above, I am missing the reference to recent studies. In the last 5 years, papers with the following numbers have been published  in addition to the papers already cited by the authors, of course,

doi: 10.3390/nu13114030, 10.1016/j.appet.2022.106300. epub 2022 Sep 6.

 doi: 10.1017/S0007114520003840. Epub 2020 Sep 30.

doi: 10.1016/j.jneb.2019.10.003. epub 2019 Nov 15.

doi: 10.1007/s40519-020-01054-8.

doi: 10.3390/nu13072388.

Please be free to read them and use them to advance the discussion.

Conclusions are relevant to the discussion and can remain as they are, even if the discussion changes.

References

As I have written previously, although the choice of references is fully appropriate to the subject matter, it is overwhelmingly literature from the last five years. However, it is suggested that this should be retained as it is extremely important in clarifying the issues discussed. It should be supplemented only by a few articles from the more recent years.

The references themselves should be adapted to the needs of the editors.

References should be in accordance with the following model

Journal Articles:

1. Author 1, A.B.; Author 2, C.D. Title of the article. Abbreviated Journal Name Year, Volume, page range.

I believe that the article presents the important issue of eating competence, which is associated with appropriate behaviours in the context of nutritional health, such as greater weight satisfaction and lower incidence of eating disorder behaviours, and has been prepared with integrity, making it a valuable addition to existing knowledge. I therefore recommend its publication, with minor corrections, in IJERPH.

Yours sincerely

Author Response

Reviewer #3:

Dear Authors,

 Thank you for giving me the opportunity to review this manuscript entitled: "Eating Competence Among Brazilian College Students", to be considered for publication at IJERPH. Below you will find my comments and suggestions, which I believe can improve the quality of the final document. I really find this manuscript interesting?

R: Thank you for your effort in helping us to improve the manuscript!

Abstract: The information is accurate. Perhaps less methodological detail could have been provided in this section. In the Reviewer's opinion, not all of the information placed on lines 18-22 is relevant at this stage.

I would suggest a rewording of the text:

Instead of: This cross-sectional study was performed using an online survey spread by snow-ball method and the instrument was applied nationwide through the Qualtrics survey software  from August 31, 2022 to November 4, 2022. A self-reported instrument (consisting of 37 questions) was divided into three parts: (i) Socioeconomic and demographic data; (ii) health data; and (iii) the  ecSI2.0™BR. The recruitment occurred through social networks and 593 college students from public and private Universities in all five Brazilian regions participated in the study. Rather: This cross-sectional study was conducted using an online survey spread using the snow-ball method. The self-report instrument was divided into three parts.

Recruitment took place through social networks and 593 students from public and private universities from all five Brazilian regions of Brazil participated in the survey.

Details removed should be included in the Materials and Methods section

R: Thank you for your comment. It was adjusted accordingly.

Introduction: The introduction provides a solid study necessary to understand the subject. However, I think the authors should have cited more recent publications from the last five or even three years on eating competence. In fact, of the 28 citations, only 7 are from the last five years. This seems all the more surprising as the adaptation for the evaluation of the Satter method has only taken place for the Spanish-speaking countries, including Brazil in the year 2020.

R: Thank you for your comment. The references were reviewed and updated.

Materials and method:

This section has been very well prepared by the authors. Nevertheless, I have a few comments.

Lines 103, please add information from abstract ”This cross-sectional study was performed using an online survey spread by snow-ball method and the instrument was applied nationwide through the Qualtrics survey software  from August 31, 2022 to November 4, 2022. A self-reported instrument (consisting of 37 questions)  was divided into three parts: (i) Socioeconomic and demographic data; (ii) health data; and (iii) the  ecSI2.0™BR”.

R: Thank you for your comment. We adjusted it accordingly.

Line 104 -  In this section the number of participants (425) differs from the abstract and results (593), please correct.

R: We apologize if we were not clear. The minimum sample size calculated was 425, but we achieved 593 participants. We modified the sentence to turn it more clear.

Results: Nothing to say here. The information is accurate

R: Thank you for your comment.

Disscusion: The discussion was properly conducted, however, following on from comments made earlier, I miss the reference to recent research. In the last 5 years, in addition to the papers already cited by the authors of course, papers have been published with the following doi numbers; doi: 10.3390/nu12010104., doi: 10.3390/nu13114030. - diabetes,Predictors of nutritional competence such as stress can also be referred to doi: The discussion has been conducted properly, but with reference to the comments above, I am missing the reference to recent studies. In the last 5 years, papers with the following numbers have been published  in addition to the papers already cited by the authors, of course,

doi: 10.3390/nu13114030, 10.1016/j.appet.2022.106300. epub 2022 Sep 6.

 doi: 10.1017/S0007114520003840. Epub 2020 Sep 30.

doi: 10.1016/j.jneb.2019.10.003. epub 2019 Nov 15.

doi: 10.1007/s40519-020-01054-8.

doi: 10.3390/nu13072388.

Please be free to read them and use them to advance the discussion.

 R: Thank you for suggestions. These references have been added.

Conclusions are relevant to the discussion and can remain as they are, even if the discussion changes.

References

As I have written previously, although the choice of references is fully appropriate to the subject matter, it is overwhelmingly literature from the last five years. However, it is suggested that this should be retained as it is extremely important in clarifying the issues discussed. It should be supplemented only by a few articles from the more recent years.

The references themselves should be adapted to the needs of the editors.

References should be in accordance with the following model

Journal Articles:

  1. Author 1, A.B.; Author 2, C.D. Title of the article. Abbreviated Journal NameYear, Volume, page range.

I believe that the article presents the important issue of eating competence, which is associated with appropriate behaviours in the context of nutritional health, such as greater weight satisfaction and lower incidence of eating disorder behaviours, and has been prepared with integrity, making it a valuable addition to existing knowledge. I therefore recommend its publication, with minor corrections, in IJERPH.

Yours sincerely